# Influence of Psychological Factors on the Success of the Ultra-Trail Runner

**DOI:** 10.3390/ijerph18052704

**Published:** 2021-03-08

**Authors:** David Méndez-Alonso, Jose Antonio Prieto-Saborit, Jose Ramón Bahamonde, Estíbaliz Jiménez-Arberás

**Affiliations:** Faculty Padre Ossó, University of Oviedo, 33008 Oviedo, Spain; josea@facultadpadreosso.es (J.A.P.-S.); jramon@facultadpadreosso.es (J.R.B.); estibaliz@facultadpadreosso.es (E.J.-A.)

**Keywords:** ultra-marathon, trail running, mental toughness, resilience, passion

## Abstract

The aim of this study was to analyze the psychological variables of runners of ultra-trail mountain races and their association with athletic performance and success. The sample was made up of 356 mountain runners, 86.7% men and 13.2% women, with a mean age of 42.7 years and 5.7 years of experience. Using pre- and post-race questionnaires, data were collected regarding mental toughness, resilience, and passion. The performance of each runner in the race was also recorded. The results showed very high values in the psychological variables analyzed compared with other sports disciplines. Completion of the race (not withdrawing) and the elite quality of the runners were presented as the most relevant indicators in the processes of resilience, mental toughness, and obsessive passion. Differences were noted between the pre- and post-race results, suggesting that the competition itself is a means of training those psychological factors that are essential to this sports discipline. It can be concluded that psychological factors are decisive to athletic performance and race completion in mountain ultra-marathon races.

## 1. Introduction

At the finish line of any ultra-marathon race, it is common to hear participants saying things such as “*… the final km. the legs just stopped working and only my head got me over the finish line,*” or “*… I was able to finish this race because I’m psychologically fit.*” This study seeks to take an in-depth look at and analyze the effect of three psychological variables on running mountain ultra-marathon races, these variables being mental toughness, resilience, and passion.

The existence of ultra-trail races has increased exponentially in recent years [1,2]. Consequently, interest in researching decisive factors to performance in these types of trials has become the focus of multiple research groups around the world. Ultra-endurance races are a multifactorial event that include physiological, neuromuscular, biomechanical, and psychological factors [3]. Multiple studies have focused on analyzing the physiological variables for improving performance in ultra-marathons [4], yet a lack of knowledge still abounds regarding the psychological factors that are unique to these runners, despite an increase in studies in recent years due to the rising popularity of these types of races [5,6]. In this sense, multiple mixed methods research projects have approached [7,8] the combination of physiological determinants (VO_2_ max, current economy, etc.) and psychological and motivational factors that have been shown to significantly influence the athletic performance of runners [4].

The influence of psychological factors on athletic performance in long-distance races has always been a widely-discussed topic, though very seldomly analyzed from a perspective of its impact on athletic performance. Without out a doubt, the variables that influence the psychological processes of athletes in highly challenging disciplines are many.

The lack of studies analyzing the psychological factors unique to these types of races [9] has led various groups to focus their work on analyzing said factors that manifest during these races [10]. Personality factors [11] with high extroversion traits; emotional factors with heightened emotional intelligence traits that can be associated with optimal adaptive psychological traits [12], motivational factors with heightened levels of intrinsic motivation [13,14], pain tolerance levels [15] and even moods that runners experience during the race and can noticeably interfere with their results [16].

Long-distance race events are one of the most stressful activities in which a human being can participate voluntarily [17] due to their intensity, duration, and potentially adverse weather conditions, in which sources of stress are in constant flux [5], and which require specific physical preparation and tremendous physical and psychological effort. It is also worth mentioning that, in the field of ultra-trail races, psychological factors have been found to play a role in the majority of cases of withdrawal [18]. For a high percentage of runners, finishing the race is the main objective increasing the number of runners reaching the goal [19]. Perceptions of success in these types of races differ according to the runner’s motivation when running the race [20]. On the one hand, there are elite runners with clear-cut performance-related motivation whose goals are completely related to achieving the best classification, while on the other hand is a separate yet large group of runners with a goal focused on completing the race; these latter will present more intrinsic motivation related to reaching the finish line [21].

Gucciardi et al. [22] define mental toughness as “a personal capacity to produce consistently high levels of subjective (e.g., personal goals or strivings) or objective performance (e.g., sales, race time, GPA) despite everyday challenges and stressors as well as significant adversities” (p. 28). Mental toughness has received considerable attention in sports as a key factor in goal achievement in the presence of various degrees of pressure, adversity, or obstacles [23]. In ultra-marathon races, mental toughness presents as a factor associated with athletic performance [24], in the same way that previous research has concluded that mental tenacity is a key factor to success in various sports disciplines [25,26,27]. Crust & Clough [28] showed evidence that the components of mental toughness are higher in individuals who can endure more extended periods of physical effort; however, subsequent studies have shown that mental toughness and self-efficacy were not significantly associated with ultra-marathon performance, although athletes require this to be of the necessary standard to prepare for and compete in elite ultra-marathon events [29]. Likewise, resilience has shown a positive correlation with sports performance and psychological well-being [30].

Windle [31], based on a review of 270 research articles, conceptualizes resilience as the process of negotiating, adapting to, or managing significant sources of stress through diverse internal psychosocial resources and contextual aspects that facilitate this capacity for adaptation and flexibility in such adverse situations.

A definition of resilience that is commonly used in sports emphasizes “the role of mental processes and behavior in promoting personal assets and protecting an individual from the potential effect of negative stressors” [32]. In the field of sport, various studies have highlighted the relationship between resilience, athletic performance [33,34] and psychological well-being [30]. In addition, resilience also relates to variables such as stress-recovery levels of athletes during the competition [35]. The study of resilience could represent an advance in the improvement of training planning and organization as well as in the athlete’s competitive performance.

As for passion, it represents a dual psychological factor, as it can be associated with either an obsessive state or a harmonious state. Passion is defined as “a strong inclination toward an activity that people like, that they find important, and in which they invest time and energy” [36]. Passion is a construct involved in psychological processes that appear in many fields of human activity such as physical activity and sports, the arts, leisure, or interpersonal relations [37,38,39]. In this sense, Vallerand et al. [39] found that, in the world of sports, both harmonious and obsessive passion were positive predictors for deliberate practice, which was simultaneously a positive predictor for objective performance. In addition, the results distinctly related the two passions to achievement goals and subjective well-being. Specifically, harmonious passion was a positive predictor for seeking mastery goals while obsessive passion was a positive predictor for mastery from the performance perspective [38]. Even though the two forms of passion may be an integral part of elite sports, athletes scoring high on obsessive passion may be at greater risk of developing burnout than more harmoniously passionate athletes [40]

Consequently, despite the increasing popularity of long-distance races, research into the psychological processes of ultra-trail runners is still quite scarce. In this sense, it is not only necessary to study the profile of long-distance runners, but also to analyze the influence of the race itself on the variability of psychological factors and their connection to performance.

Psychological factors do not only present as indicators associated with performance–understood as a better overall classification–, but also with the runner’s ability to finish the race. Factors such as the vitality states runners experience [41], self-efficacy, and intent to finish the race [42] are associated with the possibility of reaching the finish line.

In such long-distance races, controlling the emotional shifts that runners experience is considered important to being able to reach the finish line [12,43]; significant differences were found in runners for variables such as anger, confusion, or frustration between the start and end stages of a long-distance race. Similar results were found in cyclists after multi-day races involving accumulated significant loss of sleep [44]. Evidence indicates emotions associate with performance [45] and that athletes are more likely to try to regulate an emotion if they believe that doing so will facilitate performance. In the case of mental toughness, resilience, and passion, no studies have been found in which changes occurred as a result of the race.

## 2. Hypothesis and Objectives

The aim of the study was to identify the psychological profile of ultra-trail runners and the relationship between these factors and athletic success and other variables such as age and sex. At the same time, we were interested in discovering whether indicators undergo changes after highly demanding trials such as ultra-trail races.

Based on previous research in the sporting world suggesting the benefits of specific psychological factors for athletic performance and their connection with highly demanding physical activity, the following hypotheses were defined:

As an initial Hypothesis 1 (H1), it was believed that the variables analyzed in the runners’ psychological profiles (mental toughness, resilience, and passion index) would show higher scores than the sedentary population and athletes of other disciplines different from ultra-distance and would positively relate to gender, age, the athlete’s level, and their experience.

Regarding the second Hypothesis 2 (H2), the psychological factors analyzed were predicted to have a significant influence on performance and success in the race. It was expected that runners who finish the race or achieve a better time or rank would display higher scores in mental toughness and resilience.

Lastly, as a third Hypothesis 3 (H3), it was deemed that there would be significant differences in mental toughness, resilience, and passion in the pre-test and post-test results due to the set of experiences and the effort made.

## 3. Materials and Methods

### 3.1. Participants

Some 450 runners registered for the race. The study sample comprises 356 runners (79.1%) taken from the participants in the Travesera Integral Picos de Europa race, held in the Picos de Europa National Park in Spain, with an age range of 23 to 68 years and mean age of 42.73 ± 7.44. The mean number of years of experience running ultra-trail races was 5.7 years, with a minimum of 2 and a maximum of 16. The sample comprises 309 men (86.79%) and 47 women (13.2%). The mean ITRA performance index (International Trail Running Association tool for evaluating and comparing the speed of different trail runners around the world. This index compares the speed of each runner on a scale of 1000 points, corresponding to their performance against the world record for that distance) was 620 points with a maximum of 890 and a minimum of 525. The race organization provided the research group with a list of all the runners and their ITRA score (this has to be included in the registration form) which was used to assign numbers and start times. The age and sex percentages are similar to those seen in other international races such as the Ultra-Trail du Mont-Blanc [46]. The race is considered one of the most demanding ultra-trail races in the world, categorized with 5 ITRA points with a mountain coefficient of 14 (scale of 1-12), a distance of 75.9 km, and positive elevation gain of 7180 m. The average time to finish was 17 h and 53 min, up to a maximum of 21 h. The organization requires runners to show accredited prior experience in mountain races. In terms of the event on which the study is based, the race was classified as a Spanish Championship for mountain races by the Spanish Federation for Mountain and Climbing Sports, meaning that the participants were the most elite representation of the discipline at a national level. A total of 148 runners finished the race (41.57%), of which 133 were men (89.86%) and 15 were women (10.13%). The average time of those who completed the race was 17.11 h with a standard deviation of 3.15, a minimum of 19.32 and a maximum of 21.

### 3.2. Instruments

The tool used to gather the data was a survey that included questionnaires that requested sociodemographic and sport-related data (years of experience participating in long-distance mountain races), as well as various scales to assess exercise dependence, mental toughness, motivation, passion, and resilience. All the questionnaires were sent out in Spanish and 96% of the received responses were from Spanish speakers.

Mental toughness was evaluated using the 7-item Mental Toughness Inventory [25], in which participants respond using a Likert scale where 1 = False, 100% of the time, to 7 = True, 100% of the time.

The Spanish version of the 14-item Resilience Scale (RS-14) validated by Sánchez-Teruel and Robles-Bello (2014) [47] was used, derived from the original version [48] based on the 25-item Resilience Scale (RS-25) [49]. The former is a 14-item scale presented in a positive manner and with a Likert-style 7-point response format. The scale measures the degree of individual resilience, considered a positive personality trait that enables individuals to adapt to adverse situations. The RS-14 measures two factors: Factor I: Personal Competence (11 items, self-confidence, independence, decisiveness, inventiveness, and perseverance); Factor II: Self-acceptance and of life (3 items, adaptability, balance, flexibility, and perspective of a stable life).

The passion questionnaire created by Chamarro, Penelo, Fornieles, Oberst, Vallerand, and Fernández-Castro (2015) [50] was used to evaluate passion and comprises three subscales: Harmonious Passion, Obsessive Passion, and Passion Criteria, each of which has six items. The participants responded on a 6-point Likert scale ranging from 1 (Strongly disagree) to 7 (Strongly agree).

### 3.3. Procedure

Firstly, permission was obtained from the Ethics Committee of the research team’s university, and authorization was later requested from the race directors to administer the survey. The questionnaire was sent online using Google Forms the week prior to the race and the day after the competition, with the responses collected during the week after the race to ensure that there were no other competitions that could potentially falsify the post-race data. The responses were provided individually by each of the runners.

The survey was sent in one single block with the various scales separated and including instructions on how to fill in the questionnaire. Said instructions indicated that the responder should try to avoid any possible distractions and not stop part way through the survey; an estimated time for completing the survey was included (15 min). The responses that took more than 25 min to complete were not considered (8 runners). The participants gave their informed consent and filled in the questionnaires in an individual and voluntary manner during the weeks prior and subsequent to the race. Once the data were collected, the runners’ race times were added as well as their overall classification and whether they completed the course or not.

As none of the missing values exceeded 5% in any of the variables, this data did not influence the results obtained [47].

### 3.4. Data Analysis

The study design was descriptive, comparative, correlational and cross-sectional. Descriptive analyses were conducted with means, typical deviation, frequencies, and percentages to determine prevalence and create the sample description.

After performing the Kolmogorov-Smirnov test of normality and the Levene’s test for homogeneity of variance, it should be noted that the results obtained in both test show that the variables have a normal distribution and the variances are homogeneous, which allows us to carry out parametric statistics.

For the first hypothesis, descriptive statistics for each variable were used (means, typical deviation), as well performed using mean comparison contrast statistics (Student’s *t*-test) to make the comparison by gender. The analysis of differences between those who finish the test and those who drop outwas performed using mean comparison contrast statistics (Student’s *t*-test). To establish associations between psychological variables (resilience, mental toughess and passion) and rank, and race time, correlational analyses were carried out using the Pearson correlation coefficient.

For the second hypothesis, to analyze the incidence of psychological variables in the final result of the career, the sample was splited into quartiles according to the completion time in hours. First, a unidirectional Anova was used, observing if there are differences in the psychological variables between these groups of performance standards. Second, comparison contrast statistics (Student’s *t*-test) to make the comparison by quartiles.

For the third hypothesis, finally the comparison between the pre and post results were analyzed with comparison contrast statistics (Student’s *t*-test).

The program SPSS, version 25.0 (IBM, Armonk, NY, USA), was used to conduct the statistical analyses. For the purposes of data interpretation and analysis, the confidence level was 0.05 (*p* ≤ 0.05).

An attempt was made to reduce the effect of the type I error by assuming *p* ≤ 0.01 in the correlations. The perspective followed was frequentist versus Bayesian. An effort was made to avoid the so-called inverse probability fallacy in which 1-p is the probability that the alternative hypothesis is true [48].

## 4. Results

Table 1 present the results of the psychological profile of the ultra-trail runner in terms of mental toughness, resilience, and level of passion, as well as the existing correlations between the different variables. The runner presents high levels of mental toughness, resilience, and harmonious passion and low levels of obsessive passion. Significant correlations are observed between various psychological factors such as resilience, mental strength, and harmonious passion. In turn, a significant inverse correlation is found between resilience levels and obsessive passion. The results obtained are independent from the gender of the runner, except for those related to the harmonious passion, where women present significantly higher results than men (Table 2).

The runner’s age and experience level appear as influential elements in some of the psychological factors evaluated (Table 3). We can state that the levels of mental toughness and resilience increase with age and with years of experience in ultra-trail races. The elite quality of the runner, identified using their performance level based on ITRA points, positively correlates with various psychological factors, such that better runners display superior results in mental toughness, resilience, and obsessive passion.

Experience in these types of races manifests as an essential factor in the runner’s chances of completing the race. In Table 4, we can see the significant differences found between runners who finished the race and those who dropped out throughout the course according to their years of experience in ultra-marathon races.

In relation to Hypothesis 2, the results show how psychological factors play an important role in the runner’s possibility of success, both in terms of finishing the race and the time taken to complete the race, or what is considered the overall classification. “Finishers” present significantly higher psychological factors than those who withdraw in the middle of the race (Table 5). Significant differences can be observed in the factors of mental toughness, resilience, and harmonious passion, while those who withdrew from the race displayed higher results for obsessive passion.

Psychological factors are also associated with the time taken to complete the race and, therefore, the final classification obtained. The results show significant correlations (*p* ≤ 0.01) among psychological factors (mental toughness, resilience, and obsessive passion) and race time.

Upon establishing the comparison between quartiles according to overall classification time, the ANOVA results display (Table 6) the significant differences between the groups in the mental toughness, resilience, obsessive passion, and harmonious passion variables. Upon establishing the comparison between quartiles (Table 7), differences can be observed between the first group and the second and third groups and, likewise, between the fourth group and the second and third groups in mental toughness and resilience. This indicates to us that mental toughness and resilience are factors that, on the one hand, impact the ability to achieve a good classification and, on the other hand, present as factors that are essential to completing the race in the last group.

The results obtained in the comparisons between groups with the harmonious passion and obsessive passion variables reflect how obsessive passion is highly present in runners whose aim is to finish the race in one of the top positions, while harmonious passion strongly prevails in the groups whose goal is to complete the race.

In relation to working Hypothesis 3, Table 8 displays the results of the post-race questionnaire, showing the comparison of the pre- and post-race results with significant differences in distinct mental psychological factors between the week prior to the race and the week after.

## 5. Discussion

The aim of the study was to identify the psychological profile of ultra-trail runners and the relationship between these factors and athletic success and other variables such as age, sex, or experience. The study also aimed to find out if completing the race would lead to changes in the post-test results. The principal study findings verified the first two hypotheses proposed, highlighting mental toughness and resilience as predictive psychological factors for the success of ultra-trail runners. These results open new doors to preparation strategies for these types of races where runners tend to focus all their effort on physiological aspects. Nevertheless, psychological preparation takes on a decisive role in the achievement of performance-related goals.

The first hypothesis predicted high scores in the psychological factors studied. The descriptive analyses supported this prediction and displayed very high values in the dimensions of mental toughness and resilience in comparison with other sports disciplines and sedentary individuals [49,50], without finding differences relative to the runner’s sex. In this sense, the results coincide with those obtained by [51] and previously by [52]. The only differences in relation to sex where those found in the harmonious passion component, in which women scored higher than men, which could suggest a greater inclination towards the pursuit of mastery goals as compared to men and a greater focus on performance [20,27].

Years of experience and age also presented strong correlations with mental toughness and resilience. These findings are in line with prior research, though in different sports disciplines [53,54]. It is possible that the uncertain environment in these types of races results in each of the competitions representing a training session in and of itself. In this sense, a higher number of races would represent more mental training. Likewise, obsessive passion in the runner correlated with the exercise addiction inventory according to that proposed by [55] and interpretable based on the large number of training hours that the discipline requires. Similar results were reported by [56], who found that endurance sports present the highest risk of developing exercise addiction.

The present study also found positive correlations between mental toughness and the athlete’s level, meaning more elite runners scored higher on the MT scale, similar to the results found by [22,26,57,58] in various sports disciplines. Nevertheless, other studies did not find differences in the MT scale according to the athlete’s level [27,28]. It is possible that the type of sport makes a difference in this sense. The elite quality of the runner based on their ITRA score, as well as their experience, positively correlates with obsessive passion and addiction to practicing sports in line with that analyzed by [59] wherein obsessive passion can affect well-being and athlete performance over the long-term due to the associated strict exercise behavior.

The second hypothesis was also confirmed. Unlike the results obtained in ultra-marathon runners [29], mental toughness and resilience were revealed to be decisive factors for success in long-distance mountain races. The mental toughness and resilience factors manifested as decisive elements in achieving a better time and therefore higher classification given that the runners in the first quartile, elite, show significantly superior values than the runners in quartiles two and three. This also comes to light in the group in the last quartile (amateur runners whose only aspiration is to reach the finish line before the maximum time limit) in order to finish the race. Both factors present as essential elements for completing the race as differences are found between those who withdraw and those who finish. Brace et al. [29] did not find any relationship between mental toughness and performance in a 100-mile ultra-marathon race; however, in their study the runners had higher mental toughness than other sports. The authors suggest that the standard ultra-marathon runner must have heightened mental toughness but, once that threshold is reached, it is likely that other factors are more influential in determining elite ultra-marathon performance. Nevertheless, there are notable differences between ultra-marathon races on asphalt and those held in the mountains that could explain the discrepancies found in both studies. Height, elevation, and dirt trails hinder ultra-trail runners from maintaining a continuous or controlled pace, meaning it is much harder to know what pace could be maintained in the race. Self-sufficiency is another decisive factor in mountain races as compared with asphalt races, which are equipped with aid stations. In the mountain, the athletes themselves are required to plan out and carry their provisions, thereby entailing self-regulation as well as extra weight to be carried. Lastly, the solitary nature of the mountain is an additional component, unlike asphalt races where practically the entire race is completed among other runners and with pacemakers to set the pace; in the mountains it is common to run alone for long stretches of time. This accumulation of factors caused by the environment (mountain), provokes a level of uncertainty in ultra-trail runners that prevents adequate prior preparation based solely on physiological aspects and physical preparation.

The highly demanding and rigorous nature of preparing for mountain races often leads runners to consider that simply completing the race is a success in and of itself. In this sense, this can be seen in the significant differences between those who complete the race and those who withdraw before finishing, as well as the significant inverse correlation between the times of the best finishers. Previous studies did not find differences in the physiological or fitness level between runners who reach the finish line and those who withdraw [60], however, the findings of this study are in line with other works [41] where differences are observed between those who finish and those who withdraw in terms psychological factors and how the runners approach the race. Race “finishers” score significantly higher for harmonious passion, which may be associated with higher levels of positive feelings after a successful race. On the other hand, those who withdraw from the race present significantly higher levels for obsessive passion in connection to perceptions of burnout. These results are in line with the work carried out by [61]. In the same vein, the higher results in the obsessive passion component displayed by runners who withdrew from the race can be interpreted as being at a higher risk of developing burnout than the more harmoniously passionate athletes [40]. Therefore, the study results suggest that psychological factors decisively condition athletic success, this being understood not only through the prism of overall time and classification, but also the mere fact of completing the race.

Consequently, the findings confirming the second hypothesis bring to light the relevance that psychological factors such as mental toughness and resilience have on athletic success and performance in long-distance mountain races in a similar way as those obtained in other studies with athletes in similar disciplines [6,30].

Lastly, one of the aspects that most stood out in the present study are the significant differences found in the different variables between the pre- and post-race. These results confirm the third hypothesis presented. It is possible that the actual running of such a rigorous and demanding race is the reason behind the changes found in the levels of mental toughness, resilience, and addiction to physical exercise. In this sense, the results would confirm the hypothesis regarding the importance of the race itself as part of the training regimen. There is no doubt that the race itself acts as a functional element and training session for the psychological factors analyzed in a manner similar to that found in factors such as emotional control, mood, anger, or tension in races with similar characteristics [12,44]. Overcoming the effects of sleep deprivation [62], dealing with the pain of neuromuscular damage that occurs after so many hours of effort [63], and experiencing the mood shifts that occur over the course of the race [16] can be key factors in the increase in mental toughness and resilience found. The increases in harmonious passion can be easily understood due to the feelings and sensations felt by runners who complete these types of races and the experience as a whole which many runners consider to be a major life experience [64].

The post-race changes to the levels of the psychological factors can be interpreted from the perspective of the potential these races present as an element of training those same traits.

### Limitations and Strengths

The main study limitations were, on the one hand, using the results from a single ultra-trail race. Despite being one of the most prestigious national races and having the best national and international runners in attendance, studies that analyze a larger number of races are preferential. On the other hand, this research was focused exclusively on psychological aspects. Bearing in mind the multifactorial evidence of this sport, future research should analyze both physiological and psychological variables within the same study.

On the contrary, one of the strengths was the access to a majority of elite-level athletes in such a prestigious race. It is also worth mentioning that, in general, post-race questionnaires tend to notably downplay the participation of the sample; however, in this study, the sample did not display the same feeling.

## 6. Conclusions

Based on the results obtained in this study, we can conclude that athletes who participate in ultra-trail races present very specific psychological traits that enable them to adapt to the extremely tough conditions of the races. However, despite the fact that mental toughness, resilience, addiction, and passion form part of the standard runner model, age, experience, and the elite quality of the athlete accentuate this condition even more.

Mental strength and resilience are decisive factors in athletic success and performance in ultra-trail races. In this sense, athletic success should be considered in terms of both overall classification and race completion, the latter being the goal for a large part of the participants.

## Figures and Tables

**Table 1 ijerph-18-02704-t001:** Means, standard deviations and correlations among psychological variables.

	M (SD)	Mental Toughness	Resilience	Harmonious Passion	Obsessive Passion	α Crombach
Mental Toughness	6.84 (0.93)	1				0.87
Resilience	6.23 (0.65)	0.186 **	1			0.90
Harmonious Passion	6.31(0.83)	0.128 *	0.127 *	1		0.88
Obsessive Passion	2.93 (1.16)	−0.061	−0.177 *	0.106 *	1	0.84

* *p* ˂ 0.05 (bilateral), ** *p* ˂ 0.01 (bilateral).

**Table 2 ijerph-18-02704-t002:** Comparison of the results of the psychological variables according to gender.

	t	gl	Sig. (Bilateral)
Mental Toughness	−0.23	354	0.81
Resilience	−0.28	354	0.77
Harmonious Passion	−4.13	354	0.01
Obsessive Passion	−0.71	354	0.47

**Table 3 ijerph-18-02704-t003:** Correlations among psychological variables and years of experience/age/International Trail Running Association (ITRA) score.

	Years of Experience	Age	ITRA Score
Mental Toughness	0.30 **	0.33 **	0.50 **
Resilience	0.14 **	0.11 *	0.26 **
Harmonious Passion	−0.070	−0.188 **	−0.070
Obsessive Passion			0.22 **

* *p* ˂ 0.05 (bilateral). ** *p* < 0.01 (bilateral).

**Table 4 ijerph-18-02704-t004:** Comparison of results in relation to finishers and withdrawals among years of experience in ultra-trail races.

		*N*	M(SD)	*t*	gl	Sig.
Years of experience in Ultra-Trail Race	Finishers	148	6.10 (2.25)			
	Withdrawals	208	3.69 (2.22)	10.01	354	0.001 *

* *p* ˂ 0.001.

**Table 5 ijerph-18-02704-t005:** Comparison of psychological variables in relation to finishers and withdrawals.

		*N*	*t*	Sig.
Mental Toughness	Finishers	148		
	Withdrawals	208	4.25	0.01
Resilience	Finishers	148		
	Withdrawals	208	3.42	0.01
Obsessive Passion	Finishers	148		
	Withdrawals	208	−4.39	0.70
Harmonious Passion	Finishers	148		
	Withdrawals	208	−0.41	0.01

**Table 6 ijerph-18-02704-t006:** ANOVA comparative analysis by quartiles as a function of race time.

	Sum of Squares	gl	Quadratic Mean	F	Sig.
Mental Toughness	15,219	1	15,219	18,121	0.01
Resilience	4934	1	4934	11,745	0.01
Harmonious Passion	12,666	1	12,666	19,285	0.01
Obsessive Passion	0.226	1	0.226	0.168	0.682

**Table 7 ijerph-18-02704-t007:** Descriptive statistics and comparison of psychological variables and time race quartiles (expressed in hours).

Time Race Quartiles	Mental Toughness	Harmonious Passion	Obsessive Passion	Resilience
	M (SD)	M (SD)	M (SD)	M (SD)
10.32–15.01	6.58 (1.32)	5.47 (0.96) *	3.15 (1.09)	6.26 (0.39)
15.02–17.56	5.6 (0.96) *	5.15 (1.21) *	2.78 (1.44)	5.60 (0.81) *
17.57–19.10	5.74 (1.13) *	5.88 (0.80)	2.49 (0.92) *	6.04 (0.55) *
19.11–21.00	6.37 (1.08)	5.92 (0.85)	2.39 (1.22) *	6.18 (0.44)

** p* ˂ 0.01. Statistically significant comparison for mental toughness: first and fourth quartiles with second and third quartiles. Statistically significant comparison for harmonious passion: third quartile with first and second quartiles; fourth quartile with first and second quartiles. Statistically significant comparison for obsessive passion: first quartile with third and fourth quartiles. Statistically significant comparison for resilience: first quartile with second and third quartiles; fourth quartile with second quartile.

**Table 8 ijerph-18-02704-t008:** Comparation of pre-/post-race.

	Moment	M	SD	*t*	Sig.
Mental Toughness	Pre test	5.62	1.04456		
	Post test	6.21	0.57666	2.35	0.01
Resilience	Pre test	5.8614	0.65773		
	Post test	6.36	0.65683	2.82	0.01
Harmonious Passion	Pre test	5.7344	0.86276		
	Post test	6.0336	0.70145	3.09	0.01
Obsesive Passion	Pre test	2.9332	1.17083		
	Post test	2.9327	1.13836	−0.41	0.99

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
