# Peer review of "Influence of Psychological Factors on the Success of the Ultra-Trail Runner"

_ijerph, 2021, doi:10.3390/ijerph18052704_

Round 1

Author Response

Dear reviewer.

First of all, we would like to thank you for your thorough review of the article and for all the suggestions you have made to improve it.

We send you the new text with all the new features incorporated in red color

Point 1. I would suggest some theoretical consideration in the introduction of how mental toughness, resilience, passion, and exercise addition are suggested to enhance performance is required.

we have incorporated a greater theoretical basis regarding the selected psychological factors and their relationship to sports performance. We have eliminated, as suggested by the reviewers, the addiction to physical exercise.

Point 2: 3.2 Instruments – suggest deleting ‘ad hoc questionnaire’ and replace with text such as” Asurvey was constructed that assessed …”. I suggest that your tool used to gather the data was a ‘survey’ that included some ‘questionnaires’.
Please provide previous reported psychometric qualities of the questionnaires used (e.g.,Cronbach alphas etc) and then in the results, please provide the Cronbachs for this sample.

The wording has been modified in the text as suggested and Cronbachs has been incorporated in the first table of results.

Point 3: Tables and values throughout – there seems to be cultural variation in the use of “,” and “.” (comma and period) to denote decimal points. I am more accustomed to seeing “.” to represent a decimal point. I don’t wish to enforce my cultural norm on to this paper, but I do suggest that the authors investigate what is the usual format for this journal. While this is a small typographical issues, the implications are large, for example the M values in table 1 could be read a six thousand, or six. Values – e.g., means table 1. I would suggest that the data collected do not have the sensitivity to be reported to 4 decimal places; I would suggest that 2 dp would be more appropriate. This is an issues that should be addressed throughout the paper. P values, table 2 – personally I don’t like seeing p values of zero, as p values of zero do not exist. I appreciate that the p value you report has been rounded to zero by SPSS (ie maybe the score was .00000000123); instead I would suggest reporting the values as <.001.

we have adapted the values of the results to the suggestions made 

Point 4: design of the tables

we have modified the design of the tables by eliminating repeated data and taking into account the suggestions on table joins.

Point 5: This is the major issues - Discussion lines 261 onwards

As suggested, we have compared groups according to classification quartiles to obtain the differences between pre and post and to compare the results with those obtained by Brace et al. and we show the differences found with the aforementioned author.

We hope you like the changes and that they have significantly improved the quality of the article.

Thank you very much

Reviewer 2 Report

Overview

This study sought to examine ultra-trail runners’ psychological factors and more specifically, how they are related to athletic performance. It addresses a timely topic, which is being increasingly investigated in the sport psychology literature and, at the same time, remains relatively little understood. This study adopts an original approach, which consists of exploring the relations between different factors that are often studied separately. It also benefits from a large sample of high-level runners. Such population is sometimes challenging to recruit for studies, especially in competitive settings. The results showed that ultra-trail runners display some specific psychological traits, which are associated to athletic performance (i.e., race completion and race time) and runners’ level and years of experience in this sport. These findings support the importance of psychological factors in the success of ultra-endurance performances.

However, I have several concerns about the current version of the manuscript, which are expressed in the major comments. They can be summarized as follows: (i) the literature review is too straightforward and omits some aspects that are important to the present study, especially in terms of justification of the hypotheses, and rationale for the chosen variables, (ii) additional methodological information is required to ensure the validity of the collected data, especially regarding the conditions of the data collection, as well as some aspects of the data analysis, and (iii) the presentation of the results lacks of clarity and needs significant improvements.

In what follows I first address the major points and then suggest some minor changes.

Major comments

Introduction

Reading the introduction I had the feeling that the study was not well motivated. In particular, it would be useful if the authors could be clearer about the importance of the study. What is the contribution of the study with regards to the current state of the arts? Even if the topic was little investigated (as written on l.31-32), I think it is worth mentioning that the contributions have increased in the recent years, as supported by the systematic review by Roebuck et al. (2020) cited by the authors. Hence I would suggest to expand the literature review and especially, take into account other psychological factors that were previously investigated in ultra-trail running such as: emotional intelligence, profiles of moods states, psychosocial factors, or personality profiles. While these aspects are not exhaustive, they could be relevant to develop a stronger argument on how this study effectively contributes in the understanding of psychological processes and which gap in the literature the study addresses. Here are some references that I think could be useful:

Corrion, K., Morales, V., Bergamaschi, A., Massiera, B., Morin, J. B., & d’Arripe-Longueville, F. (2018). Psychosocial factors as predictors of dropout in ultra-trailers. PloS one

Hughes, S., Case, H. S., Stuempfle, K., and Evans, D. (2003). Personality profiles of iditasport ultra-Marathon participants. J. Appl. Sport Psychol.

Lane, A. M., & Wilson, M. (2011). Emotions and trait emotional intelligence among ultra-endurance runners. Journal of Science and Medicine in Sport

Lane, A. M., Terry, P. C., Stevens, M. J., Barney, S., and Dinsdale, S. L. (2004). Mood responses to athletic performance in extreme environments. J. Sports Sci.

McCormick A, Meijen C, Marcora S. (2015) Psychological determinants of whole-body endurance performance. Sports Medicine

Tharion, W. J., Strowman, S. R., and Rauch, T. M. (1988). Profile and changes in moods of ultramarathoners. J. Sport Exerc. Psychol.

Relatedly, as the characterization of psychological processes is per se complex, multi-factorial, and can be assessed with different methodologies, a stronger rationale for the chosen variables is needed. Indeed, I think that a reader who is unfamiliar with the ultra-endurance activity could not get a clear sense of why the chosen variables are particularly more relevant than other ones for this particular activity. A solution could be to (i) provide a clear definition of each variable (resilience and addiction are not defined for example) and (ii) to better argument to which extent these psychological factor are essential to ultra-endurance performance.

l. 75-80: I understand the interest and the legitimacy of investigating addiction. However, as non-expert on the topic of exercise addiction I would – perhaps naively ­­– conceive addiction as a clinical issue rather than a psychological factor per se. It is also difficult to see how addiction would be beneficial for performance. Could the authors expand the definition of (exercise) addiction, and which processes it includes?

l.86 (hypotheses): The hypotheses need a stronger support from the literature.

Also, while I am not fully aware of all the literature on the topic, I would have expected to see hypotheses about possible interactions between specific variables, for example mental toughness and resilience, or addiction and obsessive passion (I don’t know if such interactions have been investigated in the literature). Could it be a relevant approach to explore?

H1: What does “very high scores” mean? In reference to what? Higher than sedentary population, or a population in other types of sport?

Do you expect “high scores” even in the “negatively connoted” measures, that is, obsessive passion and addiction? Furthermore, as for passion, I suppose that one cannot have high scores in harmonious passion and obsessive passion subscales at the same time?

H2: I feel that the formulation of this sentence sounds like the authors are looking for a causal relationship between psychological factors and performance. However, as “only” correlations were investigated, I do not think that the methodological approach allows establishing such a causal link. I would suggest to rather replacing the term “influence” by “association” or “relationship”, for example.

What is the difference between “performance” and “success”? I do not see the distinction here.

H3: I do not understand this hypothesis. Some previous studies showed that other psychological variables are quite sensitive to post-race exhaustion (e.g., Lane et al., on POMS, Lahart et al., on emotion regulation, Hurdiel et al., on sleep deprivation and cognitive performances, etc). Therefore, could the authors provide arguments to explain why the measures of the chosen psychological factors would be more “stable”, given the fact that the runners’ general state is not comparable between pre- and post-race. More pragmatically, the conditions under which questionnaires are fulfilled are completely different, so why do the authors expect similar results?

Method

Participants:

If possible, the following precisions should be added for reproducibility:

  • It is mentioned on l.133 that validated translations of questionnaires were used. Were all participants Spanish-speakers?
  • 122-124: it is not clear here if the authors refer to the overall participants or those of the study sample. In any case it could be interesting to add the percentage of finishers (and/or withdrawers) in the whole race, as well as the percentage of finishers/withdrawers in the sample used for the study.
  • 3, l.109-113, about the ITRA index: while I acknowledge that it reflects the level of a runner on the basis of his/her past results, and it is a convenient way to have an overview on a runner’s level, it is not clear to me (i) how this piece of information about the runners was retrieved: did all participants have an ITRA record? Were the participant asked to give directly their ITRA ranking or was this info retrieved on the internet? And (ii) to which extent it is a reliable way to assess the elite quality of a runner (I think especially of the “good” athletes who were injured and are coming back to competition).
  • A large proportion of the total participants of the race actually participated in the study. How were they recruited? What were inclusion criteria? How many participants were excluded? For what reasons? How many withdrew from the study?

Procedure:

I have a few concerns here regarding the control of the conditions in which the questionnaires were fulfilled, especially regarding the following aspects:

- As the questionnaires were fulfilled remotely, I am wondering if (i) specific instructions were given to the participants before they started fulfilling the questionnaires (e.g., avoiding being distracted or interrupted) (ii) the researchers were present (or available online) during the questionnaires fulfillment (especially in case a participant had a question or needed help to understand an item in the questionnaire)

- Were all the questionnaires sent in one block or one by one?

- What were the validity criteria for assessing that the responses were valid? For example a target time between the beginning and the completion of the questionnaire could be an indicator (among others).

- More precisions are needed regarding the time span between pre-race and post-race measures (l.156-157). My understanding is that the time span (i.e., several weeks before and after the race) is quite large and can possibly vary from one participant to another. Did the authors verify that the participants did not run another race before and/or after the race in the study?

Instruments:

Were all the questionnaires and Likert scales translated?

Different Likert scales were used (from 5 to 7-point likert scales according to the questionnaire). More details about the questionnaire responses processing should be provided.

Data analysis

As the aim was to examine the relationship between different types of independent variables (ie., runners’ characteristics and race results) and dependent variables (i.e., the psychological measures), I wonder why the authors did not use a path analysis rather than a compilation of repeated correlations between different variables. In this sense path analysis appears as a suitable method to highlight the different relationships between the variables. Could it be a possibility to explore?

Also I think the authors should make clearer the treatment corresponded to each of the hypotheses for each steps of the data analysis.

l.16-165: Please specify what tests for normality and homogeneity of variance were used.

Many correlations were performed in an exploratory way. Did the authors consider correcting the p-value to avoid type I error?

Results

The presentation of the results lacks of clarity and needs substantial rewriting to me. Especially, I found that the compilation of tables without a clear development in the text makes the whole section difficult for a naïve reader to grasp the relevant information yielded by the results.

Did the authors have any missing values and/or removed values after participant removal for example? In that case, I think the percentage of the total data it represents should be indicated?

I think the subsections should be renamed for more clarity: for example “4.1. first hypothesis” requires the reader to go back to the end of the introduction. For better clarity I would also suggest rehearsing the hypotheses in the text.

Comments on the text:

In general, I think the text needs to be developed and report the significant results. For example, Table 2 contains much information yet it is unclear how the displayed results address the hypothesis.

For the Student test results please add the T, df and p-values (such as in l.180-182, 186-189).

l.194: “various psychological factors” should be defined

There is a mismatch between what is written in l.221-222 (post-test results) and the content of table 9 (years of experience).

Comment on the tables:

The captions of the tables are incomplete: the caption should describe the content of the table and report what type of information is contained in the rows and lines.

Correlation reports should specify if they are expressed in R or R².

Why does not Table 4 contain excessive passion? Why N=329 in the Age row?

Why are there missing factors in Tables 7 and 8?

Table 9 does not report post-race results.

Discussion

l.229-231: As already said above, the statistical analysis does not allow to establish predictive links between psychological measures and race results.

l.252-260: Please expand; there are several interesting ideas in this paragraph and that I think need to be developed.

l.279-281: The ending of this paragraph could be strengthened. I agree with the idea about preparation, but I think it should be argued to which extend mental toughness and resilience are important factors to overcome this “level of uncertainty” (l.280).

Regarding the arguments in l.268-279, it would be nice to refer to some articles, which performed “in-race” analyses, such as:

Antonini Philippe et al. (2017). The relationship between trail running withdrawals and race topography. Sports

Holt et al (2014). Exploring experiences of running an ultramarathon. The Sport Psychologist

Rochat et al. (2017). Comparison of vitality states of finishers and withdrawers in trail running: An enactive and phenomenological perspective. PLOS ONE

l.282-286: Yes, I totally agree, and it is a very special characteristic of this sport. I wonder if this point should be highlighted earlier in the introduction. Otherwise the terms “success” or “performance” remain elusive.

l.307-309: While I grasp the idea here, this sentence must be rephrased more clearly. I even think that this idea deserves a short development.

l.316-318: Some previous studies attempted to use “mixed methods” (but I agree that it is definitely a research avenue to develop):

Nikolaidis et al. (2020). Who Runs? Psychological, Physiological and Pathophysiological Aspects of Recreational Endurance Athletes. Frontiers in Psychology

Hauw et al (2017). Putting together first-and third-person approaches for sport activity analysis: The case of ultra-trail runners’ performance analysis. In Advances in Human Factors in Sports and Outdoor Recreation 

I noticed that throughout the paper the authors used different formulations, such as Psychological factors, Psychological indicators, Psychological traits, or Psychological profile, etc. Are these terms used interchangeably?

Minor comments

l.10 (abstract): instead of “success” why not use “race completion”? One can be finisher yet underperform and in this case, it is unlikely that one would perceive the performance as “successful”

l.12 (abstract): As several questionnaires were used, the plural form would be more appropriate here

l.28: the term “phenomenon” does not sound appropriate here. Could the authors rephrase this sentence?

l.37-40: “sleep deprivation” might be added?

l.46-47: a stronger rationale is needed here

l. 58-59: a stronger rationale is needed here l.60: a definition of resilience should be provided

l.75-76: a reference for this statement about addiction and sport should be added

l.84-85: “ the variability of psychological factor” could the authors be more specific?

l.85: could the authors make explicit what is meant by performance ?

l. 88: “other variables such as age and sex”, An umbrella term would be useful, for example “demographic variables, such as age and sex”

l. 94: “expected” rather than “believed”

l.11: what does “some 450 participants” mean? Is it an estimate or an accurate number?

l.109-112: A citation to the calculation of the performance indice (i.e., the ITRA webpage) should be added

Table 1 title: typo “deviations”

Table 3: the second row should be translated in English.

l.181: this sentence should be rephrased

l.186: runners’ instead of runner’ s?

l.187: “we can state” should be remove and the sentence could begin directly with “The level…”

Author Response

Dear Reviewer. First of all, I would like to thank you for the thorough review and the suggestions you have made to improve the quality and rigor of the text.
Please find attached the new text with all the additions resulting from the suggestions of both reviewers in red color.

Point 1: Introduction

We have incorporated into the text a greater theoretical basis for the choice of the psychological variables analyzed and try to justify their impact on sports performance using the bibliography provided by you.

We have eliminated relationships with exercise addiction as suggested by the two reviewers in order to focus on positive variables.

The wording of the working hypotheses has been modified in an attempt to make them more theoretically sound.

Point 2 Method:

Explanations on the language of the questionnaires, obtaining ITRA points and the process of sending questionnaires are included in the text.

A very important reason for having such a high participation in the study is that those who filled in the questionnaire are assigned an additional point to get a bib the following year. The annual demand for bibs is very high (2000 people) for the 450 runners authorized to run.

Point 3: Procedure and instruments

All suggestions or requirements made by the reviewer are incorporated into the text. 

Point 4:Results

The tables have been modified, deleting repeated information, tables have been joined to concentrate the information more and more explanatory comments have been added to try to make the results more solid. Cronbachs has been incorporated.

Point 5: Discussion

We have tried to incorporate all the suggestions made by the reviewer, linking them to the bibliographic citations provided.

Round 2

Reviewer 2 Report

First of all, I would like to thank the authors for their work in addressing my comments. I feel that the modifications make the paper more grounded in the literature, and hence the study is better justified. The information added in the method is also useful. I think that having removed exercise addiction is a wise decision (yet exercise addiction is definitely a topic that is worth investigating in future works, in relation to athletes’ training habits, for example). I also enjoyed the discussion that I found interesting and insightful.

I still have a few concerns regarding the data analysis and the results that I tried to express in the following point-by-point report.

Abstract

“Exercise addiction” should be removed from the keywords. Note that the term “exercise addiction” also still appears several times in the text.

Introduction

l.85-88: This paragraph about resilience could be merged with the next one

l.97-208: It sounds like the content of this paragraph repeats what has been said in the previous paragraph, so I would suggest to remove it.

l.228-246: After several readings of this section, I have the feelings that it “came out of nowhere”. Indeed, I do not really see the connection with the previous paragraph. In addition, this section contains dense – and interesting – information but I have trouble to find out what key ideas are conveyed and how they are articulated to each other. Could the authors be clearer about what is meant here? An alternative possibility could be to integrate this section earlier in the introduction, in the literature review, for example.

l.260: It should be specified what does “other disciplines” include. Does it also encompass other endurance sports, such as cycling, or marathon, for example? Indeed I am aware of a study that analyzed passion (among other things) in ski mountaineering and I wonder to which extend the results obtained for passion are comparable to the present study?

Antonini Philippe al. (2014). Passion for ski mountaineering and relationship quality: The mediating role of team cohesion. International Journal of Sport Psychology

l.263-266: This sentence should rephrased in a less “predictive" way, as for example: “it is expected that runners who finish the race or achieve a better time or rank would display higher scores in mental toughness and resilience.”

l.269: While one could grasp the idea here, the expression “the set of experience and effort made” is quite vague.

Method

l.345 (data analysis section): Reading the whole method-results sections, I felt that there is a “mix” between the descriptive statistics and those addressing the hypotheses, which makes these two sections rather unclear. In this sense, it would be clearer to structure the steps of data analysis in function of the hypotheses (i.e, what analyses on which variable were made to address each hypothesis?).

l.350-351: Shouldn’t it be “both tests” instead of “both questionnaires” ?

l.389-392: is “race” results meant here? If it is, could it be specified whether it includes finish/withdraw, rank, and/or race time?

I see that only resilience is mentioned. What comparisons were done for mental toughness?

As the normality tests reported a normal distribution, I do not understand why the authors performed a non-parametric test (i.e., the Mann-Whitney U test), which makes me reiterate the need of a more step-by-step and variable-specific description of the data analysis.

l.395-398: I do not understand this paragraph, especially because it is unclear what correction was actually made by the authors.

The analyses that yielded the results in Tables 6 and 7 are not explained in the method section.

Results

l.400: Remove the “first hypothesis” title

l.401-403: the main results reported in table 1 must be developed in the text.

Table 1: The row “obsessive passion” is missing.

For better clarity, I suggest that the row “M(SD)” could be moved to the left, in order to appear before the correlation coefficients.

The different significance thresholds expressed by the asterisks must be detailed in the table caption (as done in Table 3 for example)

l.406-414 (Tables 2 and 3): While I understand that the purpose of the investigation of gender and age is descriptive, it is unclear to which extend age and gender are relevant variables to address the hypothesis 1. One possibility would be to remove these analyses or to provide a stronger rationale for taking them into account.

Tables 4, 5 and 7: why was a non-parametric test used? If I am not mistaken, a Student t-test was used in the previous version of the manuscript.

l.629-641: classifying time performance in quartiles is a good idea and seems to be a good indicator of the runners’ performance. This step must be explained in the method.

Table 6: I guess that the race times in the quartiles are expressed in hours?

A caption is needed for this table: What do the numbers in exponent refer to?

The results of the ANOVA analysis (F and post hocs) must be reported too.

l. 642-645: this paragraph sounds more of a discussion than a report of the results 

Author Response

we upload a word file a point-by-point response to the reviewer’s comments
